# Exploring perceptions, challenges, and opportunities of UDS health professionals' students and faculty regarding community-based education programs for interprofessional education: A study protocol

**Maxwell Ateni Assibi**[1]*, **Bruce Ayabilla Abugri**[1], **Patience Afua Adwaapa Karikari**[1], **Shamsu-Deen Ziblim**[2], **Victor Mogre**[1]

**1** Department of Health Professions Education and Innovative Learning, University for Development Studies, **2** Department of Population and Reproductive Health, school of Public Health, University for Development Studies

* assibimaxwell93@gmail.com

## Abstract

### Background

Interprofessional education (IPE) is widely recognized as one of the critical approaches to ensuring that health professionals work together effectively to improve health outcomes. While community-based education (CBE) programmes have emerged as one of the promising approaches for IPE, there is little empirical evidence on how these programmes are received by students and faculty. At the University for Development Studies (UDS), the Third Trimester Field Practical Programme (TTFPP) and the Community-Based Education and Service (COBES) programme are some of the CBE programmes that offer student's experiential learning.

### Aim

This study aims to explore health professions students' and faculty members' perceptions, challenges, and opportunities associated with CBE programmes at UDS as platforms for IPE.

### Methods

The research design for this study will be qualitative in nature and underpinned by an interpretive phenomenological approach. One-on-one interviews with purposively selected final-year health professions students and faculty members who have experienced TTFPP or COBES will be conducted. The data collected for the study will be analyzed using reflexive thematic analysis with the aid of qualitative data management software. Reflexivity, peer debriefing, and member checking will be some of the strategies used to ensure the research is of high methodological rigor.

**Data availability statement:** No datasets were generated or analysed during the current study. This manuscript presents a qualitative research protocol and does not report results or pilot data at this stage. All relevant data from this study will be made available upon completion of the research. Following data collection and analysis, deidentified transcripts and thematic summaries will be stored in a secure institutional repository at the University for Development Studies (UDS), Ghana. Access to these materials will be provided in accordance with ethical guidelines and participant confidentiality agreements. Upon publication of the full study results, deidentified data—such as anonymized interview transcripts, coding frameworks, and supporting qualitative data-sets—will be made publicly available through the UDS Research Repository or other recognized open-access repositories such as Zenodo or Figshare. The repository link, DOI, and access instructions will be updated in the final published article and supporting information. Any researchers seeking access to confidential data prior to public release may direct their request to the Institutional Review Board (IRB) of the University for Development Studies at uds-irb@uds.edu.gh. Access may be granted to researchers who meet the criteria for access to sensitive data, including having ethical approval from a recognized research body and a commitment to protect participant anonymity. All data shared will comply with the requirements of the UDS-IRB, and no identifiable personal information will be disclosed. Data will be retained and preserved in accordance with UDS institutional policy and Ghanaian national data protection regulations.

**Funding:** The author(s) received no specific funding for this work.

**Competing interests:** I have read the journal's policy, and on behalf of all authors, I confirm that the authors of this manuscript have the following competing interests to declare: We recognize the importance of transparency in disclosing any potential competing interests that may influence the conduct or reporting of research. This manuscript—titled "Exploring Perceptions, Challenges, and Opportunities of UDS Health Professionals' Students and

## Expected outcomes

The research is expected to produce contextualized understandings about the way in which CBE programmes are currently experienced in relation to IPE, including facilitators, constraints, and opportunities for more structured IPE.

## 1.0 Introduction

Interprofessional Education (IPE) is described as a situation where learners from two or more professions learn about, from, and with each other to achieve an effective collaboration that enhances health outcomes [19]. IPE is a conceptual distinction from other forms of learning that may be described as interdisciplinary or multidisciplinary learning among learners from different disciplines working alongside one another. In the context of the present study, IPE is described as an educational approach that is designed to achieve the following competencies among learners: communication, role clarification, teamwork, mutual respect, and shared decision-making [13,18]. Globally, IPE has been recognized as a crucial approach to enhance health systems and collaboration among the workforce to improve patient-focused health outcomes [13,18]. Studies have shown that IPE is an effective educational approach that enhances learners' preparedness for collaboration and improves health service delivery [5,7]. Nevertheless, the effectiveness of IPE is highly contextual and is subject to the institutional culture, preparedness of the faculty members, and the availability of resources [16,18].

### 1.1 Community-based education (CBE) as a platform for IPE

Community-based education (CBE) is an experiential approach to education in which learning occurs in real-life community settings with students able to interact with real-life health and community challenges with a view to integrating theoretical knowledge with practice [5]. CBE has been identified as a powerful tool for facilitating IPE as it allows for collaborative learning, exposure to social determinants of health, as well as collaborative interactions [8,17].

At the University for Development Studies (UDS), community engagement has been institutionalized through two main programmes: The Third Trimester Field Practical Programme (TTFPP) and Community-Based Education and service (COBES) programme. TTFPP is a UDS -wide initiative in which students across disciplines, including health-related ones, are placed in community settings to undertake development projects [9]. Although TTFPP has a multidisciplinary composition, it has the potential for IPE. COBES, on the other hand, is a mono-professional programme for medical students only in UDS. However, with a focus on community immersion, service learning, and reflection, it has a high possibility for teamwork engagement in future.

Notably, the coexistence of various professions in the programme does not, in itself, amount to IPE. There is need to ensure planned educational design, facilitation, and alignment with IPE competency frameworks to ensure exposure to

Faculty Regarding Community-Based Education Programs for Interprofessional Education"—is an independent academic protocol prepared by the authors as part of postgraduate training in health professions education at the University for Development Studies (UDS), Tamale, Ghana. The study is entirely self-funded. No grants, donations, or external financial contributions were received from public agencies, commercial enterprises, or private foundations during the development of the protocol. None of the authors have received salaries, honoraria, consultancy fees, or any form of monetary compensation related to this research project. The study has not been commissioned by any institution or third party with vested interest in its outcome. No author holds stock or shares in organizations that may benefit from this research, nor do any of the authors serve as board members, officers, or employees of such institutions. None of the authors are engaged in lobbying, advocacy, or advisory roles in organizations that support or oppose community-based education (CBE) or interprofessional education (IPE) initiatives. Regarding non-financial competing interests: none of the authors are aware of any personal, political, academic, or intellectual interests that could be perceived to affect the neutrality, objectivity, or integrity of the work presented. Though some authors are affiliated with departments responsible for implementing the COBES and TTFPP programs at UDS, this research is conducted independently with no direct influence from administrative authorities. Measures will be taken throughout the research process to mitigate potential bias, including triangulation, member checking, peer debriefing, and researcher reflexivity practices. The corresponding author, Maxwell Ateni Assibi, is a final-year Master of Health Professions Education (MHPE) student conducting this research as part of the thesis requirement under faculty supervision. The faculty co-authors involved in this work contributed in their capacity as academic advisors and reviewers of the protocol and are not in positions of administrative power over program policy or student assessments related to COBES or TTFPP. The research has received no material support or endorsements from governmental health departments, non-governmental organizations (NGOs), or donor agencies involved in healthcare or higher education policy within Ghana or abroad. Neither the study's conception, methodology, nor expected dissemination

interdisciplinary results in meaningful IPE [6,13]. This is the basis on which the conceptual framework of the current study is founded.

## 1.2 Research gap and rationale

Whilst CBE has been explored in several contexts of high-income countries and some middle-income countries [5,17], there is a scarcity of empirical research that investigates the perceptions of existing CBE programmes in low-resource contexts as a basis for IPE. Specifically, there is a lack of understanding of how students and faculty perceive the potential for IPE between structurally distinct programmes, such as mono-professional or multidisciplinary CBE programmes.

With regards to the institutional context, CBE programmes have been established at the UDS for several decades. However, there has been a scarcity of systematic qualitative research that has examined the current role of CBE programmes in supporting or limiting the potential for IPE. The absence of such research has significant implications for curriculum development initiatives, faculty development programmes, and policy alignment.

This is particularly significant for UDS and several institutions in sub-Saharan Africa that may want to capitalize on existing educational infrastructures to support the advancement of IPE. The perceptions of stakeholders constitute a critical first step in this process.

## 1.3 Problem statement

IPE has been identified as a fundamental framework for the education of health professionals to work together effectively in a complex health system [19]. Yet the implementation of IPE in the curricula of health professions has been inconsistent, especially in resource-poor countries [16].

The UDS offers a number of CBE programmes that provide students with the opportunity for experiential learning in a community-based context. TTFPP is an example of an educational programme that involves students from a number of disciplines, while COBES is a mono-professional programme that only involves medical students. The long history of implementation of these programmes makes it difficult to determine whether they now promote IPE or only provide students with an interdisciplinary experience or simply have the potential for the integration of IPE.

There is limited empirical qualitative evidence examining how health professions students and faculty interpret existing CBE programmes in relation to IPE, including the challenges that constrain collaborative learning and the opportunities for deliberate curricular integration of IPE. In the absence of such context-specific insight, initiatives aimed at strengthening IPE within CBE risk conceptual and implementation misalignment. This therefore underscores the need to explore the perceptions, challenges, opportunities, and enabling conditions that shape the integration of interprofessional education within existing community-based education programmes.

## 1.4 Aim of the study

The aim of this study is to explore the perceptions, challenges, and opportunities associated with CBE programmes as platforms for promoting IPE.

is intended to promote the interests of any particular stakeholder or institutional agenda. Furthermore, there are no pending patents, patent applications, or proprietary technologies arising from this study. The protocol does not involve testing or evaluation of any medical products, interventions, or services that could result in commercial gain or product development. All authors have agreed to the contents of this statement and confirm that they have nothing to hide that could constitute a financial or non-financial competing interest in relation to the content of this manuscript. Should any future conflict of interest arise during the study execution or publication phase, the authors pledge to immediately notify the journal editorial team and update the disclosure accordingly. In conclusion, the authors have declared that no competing interests exist, financial or otherwise, that could be perceived to influence this study. This declaration is made in good faith and in alignment with the PLOS ONE editorial and ethical standards for responsible conduct of research.

## 1.5 Research questions

1. How do health professions students and faculty at UDS perceive CBE programmes in relation to IPE?

2. What challenges do students and faculty identify in utilizing TTFPP and COBES as platforms for IPE?

3. What opportunities do students and faculty perceive for strengthening IPE within existing CBE programmes at UDS?

## 1.6 Objectives of the study

**1.6.1 General objective.** To explore perceptions, challenges, and opportunities related to the use of CBE programmes for promoting IPE at the UDS.

**1.6.2 Specific objectives.**

1. To examine health professions students' and faculty members' perceptions of CBE programmes in fostering IPE competencies such as communication, teamwork, and role clarification.

2. To identify perceived structural, pedagogical, and institutional challenges limiting the integration of IPE within TTFPP and COBES.

3. To explore perceived opportunities for strengthening and intentionally integrating IPE within existing CBE programmes.

## 1.7 Significance of the study

This study will contribute to the body of research on IPE by adding qualitative research findings that highlight the experience of CBE programmes in a low-resource university context. By differentiating between interdisciplinary exposure and structured interprofessional education (IPE), the study aims to provide conceptual clarity that can inform both research and educational practice. The study's findings may also contribute to the body of knowledge for other health professions education institutions seeking to capitalize on existing CBE programmes to improve collaborative practice in resource-constrained contexts.

## 2.0 Methodology

This study protocol is reported in accordance with the Standards for Reporting Qualitative Research (SRQR) guidelines [11].

## 2.1 Study design and research paradigm

The research will be guided by a qualitative research paradigm with an underlying interpretive phenomenological approach [3,10]. This approach is appropriate for this study as it seeks to explore the way in which the experiences of the students in UDS are perceived as enabling, constraining, or potentially supporting IPE by the students and faculty. Unlike quantitative research, this approach will allow for

the in-depth exploration of the realities of the participants' perspectives, interpretations, and understandings within the UDS.

The research will be guided by an interpretivist paradigm, which asserts that reality is socially constructed through interaction, experience, and context [4,15]. This paradigm is appropriate for research as IPE is a concept that is inherently relational, based on the interactions of different people in different settings.

## 2.2 Study setting

The proposed study will be conducted within the context of UDS, located in the campus in Tamale, Ghana. UDS is a public university with an institutional mandate that prioritizes CBE. Two major CBE programmes that the university offers and which are pertinent to this proposed study include TTFPP, which is involved from students from diverse academic fields, and COBES, which involves only medical students.

## 2.3 Study population

The study population will comprise:

1. Final-year health professions students enrolled in medicine, nursing, midwifery, pharmacy, and nutrition who have completed either TTFPP or COBES.

2. Faculty members who have directly supervised, coordinated, or taught within TTFPP or COBES.

The students in their final year are chosen because they have gone through the entire process of learning in the community setting. The faculty is chosen because they are involved in the design, supervision, and facilitation of the learning process.

## 2.4 Inclusion and exclusion criteria

### 2.4.1 Inclusion criteria:.

1. Final-year students who have completed TTFPP or COBES

2. Faculty with direct involvement in the supervision or coordination of TTFPP or COBES

3. Willingness to provide informed consent

### 2.4.2 Exclusion criteria:.

1. Students or faculty unavailable during the data collection period

2. Individuals who decline consent

Participants are not excluded based on health status; rather, non-participation will be managed through flexible scheduling to minimize selection bias.

## 2.5 Sampling strategy and sample size

A purposive sampling approach will be used [12] to select participants with direct experience related to research questions. The sampling will target maximum variation based on profession and role to ensure a range of perspectives.

The proposed sample will consist of:

1. Around 20 final-year students, with a range of health professions

2. Around 10 faculty members, from a variety of health disciplines and roles

Sampling will proceed until a point of data saturation is reached [14], where no further conceptual insights are gained.

## 2.6 Recruitment procedures

This will be carried out by the principal investigator, working in collaboration with the administrators of the various departments. Information sheets will be disseminated through the departments, and the respondents will be requested to contact the team directly if they wish to participate. This strategy avoids coercion and maintains the voluntariness of the respondents.

## 2.7 Data collection

The data collection methods will include conducting semi-structured one-on-one interviews [3].

Interview guides will be developed separately for students and faculty to address their different roles. The interviews will be conducted by the principal investigator. The principal investigator is trained in conducting interviews.

The interview guide will have open-ended questions such as:

1. "Can you describe your experiences of working with students or professionals from other disciplines during community-based placements?"

2. "In what ways, if any, do you think TTFPP and COBES supports or limits IPE?"

3. "What opportunities do you see for strengthening IPE within these programmes?"

The interviews will take approximately 30–45 minutes and will be audio-recorded.

## 2.8 Piloting

The interview guide will be piloted with 2–3 participants who have similar characteristics to those in the population. This will ensure methodological rigor rather than notions of validity and reliability.

## 2.9 Data management and storage

All audio recordings will be transcribed literally by qualified research assistants under a confidentiality agreement. The data will be anonymized by removing any identifiable information. The data will be stored on computers that are password-protected, as well as encrypted storage media that is accessible only to the research team. The audio files will be erased once the transcription is complete.

## 2.10 Data analysis

Data will be analyzed using reflexive thematic analysis, following the procedures outlined by Braun and Clarke [2]. NVivo (Version 18.0) will be used to support data organization and coding.

The analysis will proceed in several steps.

First, all transcripts will be read several times to allow familiarization with the data and the development of initial ideas. Notes will be taken during this stage to capture early impressions.

Second, initial codes will be developed from meaningful segments of the data related to the research questions. Coding will be carried out across the entire dataset.

Third, the codes will be grouped into potential themes by identifying patterns and relationships among them.

Fourth, the themes will be reviewed and refined to ensure they accurately represent the data and are clearly distinct.

Fifth, each theme will be clearly defined and named.

Finally, the findings will be presented in a narrative form, supported by selected excerpts from participants' responses.

To enhance the trustworthiness of the analysis, a second researcher will independently code a portion of the transcripts, and any differences will be discussed and resolved.

## 2.11 Trustworthiness and rigor

Trustworthiness will be established through multiple complementary strategies grounded in accepted qualitative standards. The researcher will maintain a structured reflexive journal throughout data collection and analysis to document methodological decisions, evolving assumptions, and potential sources of bias, thereby strengthening confirmability. Regular peer debriefing with supervisors and co-authors will involve systematic scrutiny of coding processes, theme development, and interpretive claims to enhance analytic credibility. Member checking will be conducted by providing participants with concise summaries of the researchers' interpretations to verify accuracy and ensure faithful representation of their perspectives. In addition, a comprehensive audit trail comprising raw data, coding schemes, analytic memos, and records of methodological decisions will be maintained to support dependability and transparency. Collectively, these procedures will enhance credibility, dependability, confirmability, and transferability of the findings.

## 2.12 Researcher reflexivity and positionality

The principal investigator is an educator in health professions and has an association with UDS [1]. It is important to be aware of this, especially considering the potential power dynamics and pre-existing assumptions. Reflexive journaling, peer debriefing, and documentation will be employed to reduce the impact of the investigator on the data collection and interpretation process.

## 2.13 Ethical considerations

This investigation has obtained its ethical approval from the Institutional Review Board of the UDS (Ref. no: UDS/RB/0178/25). Written informed consent shall be sought from each participant before the commencement of data collection. The study's participation will be completely voluntary, and the participants will be allowed to exercise their right to withdraw at any time during the study without facing any negative consequences.

Confidentiality and anonymity will be maintained through the entire research process. Personal information will be taken out of all transcripts and datasets, and participants will be given individual codes. All the data will be protected and saved on password-secured computers as well as encrypted storage devices which only the research team can access. After the transcription and verification, the audio recordings will be erased.

The research will be carried out in compliance with internationally recognized ethical standards for research with human participants.

## 3.0 Expected results/ Anticipated outcomes

In as much as this paper is a protocol for a study, the anticipated results outlined below are general and non-conclusive. The purpose of outlining them is to give a general idea of the nature of the findings that this study is intended to produce.

The anticipated findings of this study include that health professions students and faculty have generally positive perceptions of CBE programmes as learning environments. Participants may report that these programmes promote learning environments that offer exposure to teamwork, community engagement, and problem-solving. However, these learning environments may be characterized as interdisciplinary exposures with little focus on IPE competencies.

The study is anticipated to reveal a number of the perceived challenges to the integration of IPE into the existing CBE programs. These may include the lack of clear IPE objectives, the lack of coordination between academic programs, the lack of coordination in the academic calendars, and the lack of institutional guidance on IPE and teaching approaches. Faculty members may also reveal challenges related to facilitation approaches, workload, and the lack of processes to assess IPE.

Furthermore, the study is anticipated to reveal the perceived opportunities to improve IPE and teaching in the CBE programs at UDS. These may include the addition of IPE and teaching approaches, the addition of learning objectives, the addition of reflective learning approaches, and the addition of other approaches

## 4.0  Potential implications

Although this study will not produce empirical findings in its protocol stage, the anticipated insights from the perspectives of students and faculty members may contribute to curriculum development in health professions education. The understanding of how CBE programmes are viewed in relation to IPE may contribute to the alignment of learning, teaching, and assessment strategies with essential competencies such as communication, teamwork, and role clarification, in order to structurally integrate IPE activities into existing CBE programmes.

The anticipated study findings may also contribute to faculty development and policy alignment in health professions education. The understanding of perceived facilitators and barriers to IPE may lead to an understanding of where faculty development is needed to effectively facilitate IPE. The study may also contribute to evidence-based decisions in health professions education, such as faculty development and policy alignment, in relation to cross-departmental collaboration, scheduling, and supervision. Furthermore, it may contribute to broader policy discussions in relation to strengthening IPE in low-resource settings through the utilization of existing CBE programmes.

## 5.0  Protocol summary

The following is a study protocol for a qualitative exploration of health professions students' and faculty members' perspectives on CBE programs as a platform for IPE. In addition to this, it provides an outline of the conceptual framework of the study, methodology, and analysis to ensure transparency, rigor, and reproducibility of the research findings. This study seeks to provide foundational evidence for future research in this area.

## Author contributions

**Conceptualization:** Maxwell Ateni Assibi.

**Supervision:** Shamsu-Deen Ziblim, Victor Mogre.

**Writing – review & editing:** Bruce Ayabilla Abugri, Patience Afua Adwaapa Karikari.

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
