## [Decision Letter · Decision Letter 0]

14 Jan 2026

Dear Dr. Assibi,

Thank you for submitting your manuscript to PLOS ONE. After careful consideration, we feel that it has merit but does not fully meet PLOS ONE’s publication criteria as it currently stands. Therefore, we invite you to submit a revised version of the manuscript that addresses the points raised during the review process.

We look forward to receiving your revised manuscript.

Kind regards,

Saravana Kumar

Academic Editor

PLOS One

Journal Requirements:

“This research is self-funded. No external financial support was received for the design, data collection, analysis, or writing of this protocol.”

6. Please amend either the title on the online submission form (via Edit Submission) or the title in the manuscript so that they are identical.

Reviewers' comments:

Reviewer's Responses to Questions

**Comments to the Author**

1. Does the manuscript provide a valid rationale for the proposed study, with clearly identified and justified research questions?

Reviewer #1: Partly

Reviewer #2: Yes

2. Is the protocol technically sound and planned in a manner that will lead to a meaningful outcome and allow testing the stated hypotheses?

Reviewer #1: Partly

Reviewer #2: Yes

3. Is the methodology feasible and described in sufficient detail to allow the work to be replicable?

Reviewer #1: Yes

Reviewer #2: Yes

4. Have the authors described where all data underlying the findings will be made available when the study is complete?

The PLOS Data policy requires authors to make all data underlying the findings described in their manuscript fully available without restriction, with rare exception, at the time of publication. The data should be provided as part of the manuscript or its supporting information, or deposited to a public repository. For example, in addition to summary statistics, the data points behind means, medians and variance measures should be available. If there are restrictions on publicly sharing data—e.g. participant privacy or use of data from a third party—those must be specified.requires authors to make all data underlying the findings described in their manuscript fully available without restriction, with rare exception, at the time of publication. The data should be provided as part of the manuscript or its supporting information, or deposited to a public repository. For example, in addition to summary statistics, the data points behind means, medians and variance measures should be available. If there are restrictions on publicly sharing data—e.g. participant privacy or use of data from a third party—those must be specified.

Reviewer #1: No

Reviewer #2: Yes

5. Is the manuscript presented in an intelligible fashion and written in standard English?

Reviewer #1: Yes

Reviewer #2: Yes

You may also provide optional suggestions and comments to authors that they might find helpful in planning their study.

Reviewer #1: This protocol addresses an important area in health professions education by proposing to examine community-based education programmes as platforms for interprofessional education. Given that protocols are intended to establish conceptual clarity, methodological coherence, and feasibility prior to study implementation, several aspects of the protocol would benefit from further clarification. Addressing these points at the protocol stage is important for ensuring methodological coherence and strengthening the study’s eventual scholarly contribution.

The following questions in the attached document are posed to support refinement of the study’s conceptual framing, and the intended contribution of the study.

Reviewer #2: Thank you for the opportunity to review this protocol. This protocol provides an informative overview of the planned study in an interesting and valuable area of research. Some key areas to review and revise in this protocol are described below:

• Please specify whether you plan to use a reporting guideline to inform the reporting of your study?

• A position statement is required to provide background on the researchers and their position in the research.

• Please review for inconsistencies in the formatting of your in-text references throughout the manuscript. For instance, including author’s initials within the in-text reference.

• Try to be consistent with your use of abbreviations e.g. Community-based education (CBE) and University of Development Studies (UDS) and avoid redefining abbreviations in each section that have already been provided earlier in the manuscript.

Please see below for feedback relevant to specific sections of your manuscript:

Abstract

• In the methods, please state the specific qualitative research methodology that will be used. Please also identify the thematic analysis process that will be undertaken.

Introduction

• Please check the first reference (WHO.2010) for an extra full stop.

• In paragraph 2, please review the reference (Green BN & Johnson CD., 2015) as authors’ initials not required here.

Aim & research questions

• You have clearly outlined the research questions and objectives in your manuscript, it would be beneficial to include some examples of the interview questions which you will use, to see how they align with the research questions and objectives you have provided.

Population of the study

• Who will be responsible for recruitment of the participants and what will this process entail?

• What sampling strategies will be used as part of this study? I can see this is mentioned in a later section, you could think about incorporating the information about sampling in this section where you describe the sample you will aim to recruit, in order to cut down on excess words and repetition later.

• As you plan to recruit students and members of the faculty, are you going to set a quota for the number of participants from each of these subgroups (i.e. use of quota sampling)?

Variables of the study

• I’m not sure if this section is necessary, considering this is a qualitative study and variables are more of a feature of quantitative study design. I would recommend thinking about your terminology here or removing this section all together.

Sample size

• You have stated that you will aim to recruit “approximately ten faculty members representing a diverse range of health professions”. Will you aim to recruit a certain number from each health profession to capture a variety of different professions within your sample?

Data collection procedures

• Who will be conducting the interviews with the participants? What is their experience in conducting qualitative interviews and will they pilot any interviews prior to data collection?

• You have mentioned the use of an interview guide; can you provide some examples of interview questions that will be included?

• Please identify who will be responsible for transcribing the interviews and how.

• Please specify whether you will conduct any follow up interviews with participants.

Pre-testing/ pilot testing

• Be careful with using terminology such as “validity and reliability” as this is more relevant to quantitative research.

• How many participants will be involved in the pilot testing of the interviews?

• Try to be consistent with the terminology used (re. pilot testing vs. pre-testing).

Data analysis

• Please add an in-text reference for the Nvivo software that you have referred to.

• You have commented on the use of peer debriefing and member checking to achieve validity and reliability. As with my previous comment, please be careful with using this terminology considering this is a qualitative study. Likewise, further explanation of the member checking process is necessary in this section (for instance, what information will be shared with participants in the “summary”?) to explain the purpose of member checking and how it supports rigour and trustworthiness.

• As part of the coding process, will you have multiple researchers involved in the coding?

• “Triangulation of data will also be done through cross-matching results in different sources of data”. What does this mean, can you explain this?

Sources of data

• Spelling error at the end of the first sentence of this section.

Reference list

• Check for inconsistencies in line spacing between each reference.

.

Reviewer #1: No

Reviewer #2: No

---

## [Author Response · Author response to Decision Letter 1]

19 Feb 2026

We have carefully revised the manuscript to address all editorial and reviewer concerns. Specifically:

The abstract has been revised to clearly specify the qualitative methodology and reflexive thematic analysis approach.

The introduction has been strengthened to improve conceptual clarity, correct referencing errors, and enhance justification of the research gap.

The aims, objectives, and research questions have been refined, and examples of interview questions have been added to demonstrate alignment.

The methodology section has been substantially expanded to improve transparency and reproducibility, including detailed descriptions of the research design, sampling, recruitment, data collection, pilot testing, data analysis, reflexivity, and trustworthiness strategies.

Reporting guidelines (SRQR) have been explicitly stated, and researcher positionality has been included.

The ethics statement has been revised to include full approval details and informed consent procedures, as requested.

The funding, data availability, and declaration sections have been corrected to comply fully with PLOS ONE policies.

Reference formatting, abbreviations, and in-text citations have been standardized throughout the manuscript.

Minor typographical and formatting issues have been corrected.

All changes have been highlighted in the tracked-changes version of the manuscript, and detailed responses to each reviewer comment have been provided in the Response to Reviewers document.

We believe that the revised manuscript now meets the journal’s requirements and addresses all reviewer and editor concerns. We are grateful for the opportunity to revise our work and respectfully submit this revised version for further consideration.

---

## [Decision Letter · Decision Letter 1]

11 Apr 2026

Dear Dr. Assibi,

Thank you for submitting your manuscript to PLOS ONE. After careful consideration, we feel that it has merit but does not fully meet PLOS ONE’s publication criteria as it currently stands. Therefore, we invite you to submit a revised version of the manuscript that addresses the points raised during the review process.

As the corresponding author, your ORCID iD is verified in the submission system and will appear in the published article. PLOS supports the use of ORCID, and we encourage all coauthors to register for an ORCID iD and use it as well. Please encourage your coauthors to verify their ORCID iD within the submission system before final acceptance, as unverified ORCID iDs will not appear in the published article. *Only* the individual author can complete the verification step; PLOS staff the individual author can complete the verification step; PLOS staff the individual author can complete the verification step; PLOS staff the individual author can complete the verification step; PLOS staff *cannot* verify ORCID iDs on behalf of authors.verify ORCID iDs on behalf of authors.verify ORCID iDs on behalf of authors.verify ORCID iDs on behalf of authors.

We look forward to receiving your revised manuscript.

Kind regards,

Saravana Kumar

Academic Editor

PLOS One

Journal Requirements:

Reviewers' comments:

Reviewer's Responses to Questions

**Comments to the Author**

1. Does the manuscript provide a valid rationale for the proposed study, with clearly identified and justified research questions?

Reviewer #1: Yes

Reviewer #2: Yes

2. Is the protocol technically sound and planned in a manner that will lead to a meaningful outcome and allow testing the stated hypotheses?

Reviewer #1: Yes

Reviewer #2: Yes

3. Is the methodology feasible and described in sufficient detail to allow the work to be replicable?

Reviewer #1: Yes

Reviewer #2: Yes

4. Have the authors described where all data underlying the findings will be made available when the study is complete?

The PLOS Data policy requires authors to make all data underlying the findings described in their manuscript fully available without restriction, with rare exception, at the time of publication. The data should be provided as part of the manuscript or its supporting information, or deposited to a public repository. For example, in addition to summary statistics, the data points behind means, medians and variance measures should be available. If there are restrictions on publicly sharing data—e.g. participant privacy or use of data from a third party—those must be specified.requires authors to make all data underlying the findings described in their manuscript fully available without restriction, with rare exception, at the time of publication. The data should be provided as part of the manuscript or its supporting information, or deposited to a public repository. For example, in addition to summary statistics, the data points behind means, medians and variance measures should be available. If there are restrictions on publicly sharing data—e.g. participant privacy or use of data from a third party—those must be specified.

Reviewer #1: Yes

Reviewer #2: Yes

5. Is the manuscript presented in an intelligible fashion and written in standard English?

Reviewer #1: Yes

Reviewer #2: Yes

You may also provide optional suggestions and comments to authors that they might find helpful in planning their study.

Reviewer #1: Dear authors,

Thank you for your responses and edits to the earlier comments. A few minor edits remain to be addressed:

Line 76. CBE must be written in full at first mention and the abbreviation in brackets i.e., Community-based education (CBE)

Line 81. Same issue as above with UDS.

Line 82. Similar issues with TTFPP and COBES

Line 158. The sentence is abstract; it does not explain how the study provides clarity.

Consider revising as follows: “By differentiating between interdisciplinary exposure and structured interprofessional education (IPE), the study aims to provide conceptual clarity that can inform both research and educational practice. The study's findings may also contribute to the body of knowledge for other health professions education institutions seeking to capitalize on existing CBE programmes to improve collaborative practice in resource-constrained contexts.”

Wishing you all the best.

Reviewer #2: Thank you for taking the time to revise your manuscript with the feedback provided. You have adequately addressed the suggestions to your manuscript. After reviewing the revised manuscript, my minor feedback on the manuscript is included below:

-You have identified a reporting guideline (SRQR) that will be followed, can you please include a reference to this checklist.

-In section 2.10 data analysis, more detailed explanation of each step of the thematic analysis process is required for repeatability, instead of just listing the steps involved.

.

Reviewer #1: No

Reviewer #2: No

---

## [Author Response · Author response to Decision Letter 2]

13 Apr 2026

we have addressed all comments in the manuscript

---

## [Editor Report · Decision Letter 2]

15 Apr 2026

Exploring Perceptions, Challenges, and Opportunities of UDS Health Professionals' Students and Faculty Regarding Community-Based Education Programs for Interprofessional Education: A study protocol.

PONE-D-25-39841R2

Dear Dr. Assibi,

We’re pleased to inform you that your manuscript has been judged scientifically suitable for publication and will be formally accepted for publication once it meets all outstanding technical requirements.

Kind regards,

Saravana Kumar

Academic Editor

PLOS One

---

## [Editor Report · Acceptance letter]

PONE-D-25-39841R2

PLOS One

Dear Dr. Assibi,

I'm pleased to inform you that your manuscript has been deemed suitable for publication in PLOS One. Congratulations! Your manuscript is now being handed over to our production team.

Kind regards,

on behalf of

Professor Saravana Kumar

Academic Editor

PLOS One